EMBO
Molecular Medicine

# Fasting-induced liver GADD45β restrains hepatic fatty acid uptake and improves metabolic health

Jessica Fuhrmeister[1], Annika Zota[1,2,3], Tjeerd P Sijmonsma[1], Oksana Seibert[1], Şahika Cıngır[1], Kathrin Schmidt[4], Nicola Vallon[1], Roldan M de Guia[1], Katharina Niopek[1,2,3], Mauricio Berriel Diaz[1,2,3], Adriano Maida[1,2,3], Matthias Blüher[5], Jürgen G Okun[4], Stephan Herzig[1,2,3,*] & Adam J Rose[1,**]

## Abstract

Recent studies have demonstrated that repeated short-term nutrient withdrawal (i.e. fasting) has pleiotropic actions to promote organismal health and longevity. Despite this, the molecular physiological mechanisms by which fasting is protective against metabolic disease are largely unknown. Here, we show that, metabolic control, particularly systemic and liver lipid metabolism, is aberrantly regulated in the fasted state in mouse models of metabolic dysfunction. Liver transcript assays between lean/healthy and obese/diabetic mice in fasted and fed states uncovered "growth arrest and DNA damage-inducible" GADD45β as a dysregulated gene transcript during fasting in several models of metabolic dysfunction including ageing, obesity/pre-diabetes and type 2 diabetes, in both mice and humans. Using whole-body knockout mice as well as liver/hepatocyte-specific gain- and loss-of-function strategies, we revealed a role for liver GADD45β in the coordination of liver fatty acid uptake, through cytoplasmic retention of FABP1, ultimately impacting obesity-driven hyperglycaemia. In summary, fasting stress-induced GADD45β represents a liver-specific molecular event promoting adaptive metabolic function.

**Keywords** FABP1; hormesis; lipid; metabolism; stress

**Subject Category** Metabolism

## Introduction

The incidence of obesity is at an epidemic level worldwide and is a strong risk factor for a number of ageing-related diseases including type 2 diabetes (T2D), cardiovascular disease and the metabolic syndrome, and thus poses a tremendous burden on quality of life and health care systems worldwide (Popkin et al, 2012). Thus, there is a desperate need for more effective strategies to curtail this trend, whether through prescription of behavioural or pharmacological treatments.

A hallmark of obesity-driven T2D is insulin resistance (Bjorntorp, 1997), and thus, much effort is placed into treatments that promote "insulin sensitisation" (Connor et al, 2015). While there is no doubt that insulin is an important mediator of metabolic control in the prandial state (Boucher et al, 2014), insulin resistance likely represents a physiological feedback mechanism to actually retard the development of obesity-driven complications (Hoehn et al, 2009), prompting speculation that "insulin sensitisation" may be a flawed strategy (Connor et al, 2015). This is exemplified by studies demonstrating that tissue-restricted loss of function of key insulin signalling nodes actually extends health span of mice (Bluher et al, 2003; Taguchi et al, 2007) and that insulin per se can promote the progression of obesity-related metabolic dysfunction (Mehran et al 2012). Thus, alternative strategies are warranted, such as mild and intermittent activation of stress-responsive pathways that are pro-adaptive (Ristow & Zarse, 2010; Kolb & Eizirik, 2012).

One such strategy could be intermittent nutrient withdrawal. Nutrient withdrawal (i.e. fasting) is protective against modern chronic diseases such as cancer, neurodegeneration, cardiovascular disease and diabetes (Longo & Mattson, 2014). In particular, a recent study demonstrated that across multiple species including humans, a periodic diet that mimics fasting had a positive effect on lifespan and biomarkers of cognitive, immune and cardiometabolic function (Brandhorst et al, 2015). Concerning metabolic function, although it is known that there are several systemic adaptive changes in metabolism with nutrient withdrawal (Cahill, 2006), little is known of the tissue-specific molecular mechanisms by which fasting can improve organismal metabolic control and retard the development of metabolic disease (Longo & Mattson, 2014).

Although it is well accepted that the liver is a pivotal organ contributing to metabolic control (van den Berghe, 1991), whether

1   Joint Research Division Molecular Metabolic Control, German Cancer Research Center, Center for Molecular Biology, Heidelberg University and Heidelberg University Hospital, Heidelberg, Germany
2   Institute for Diabetes and Cancer (IDC), Helmholtz Center Munich, Neuherberg, Germany
3   Joint Heidelberg-IDC Translational Diabetes Program, Inner Medicine I, Heidelberg University Hospital, Neuherberg, Germany
4   Division of Inherited Metabolic Diseases, University Children's Hospital, Heidelberg, Germany
5   Department of Medicine, University of Leipzig, Leipzig, Germany
    *Corresponding author. Tel: +49 89 3187 1045; E-mail: stephan.herzig@helmholtz-muenchen.de
    **Corresponding author. Tel: +49 6221 423588; Fax: +49 6221 423595; E-mail: a.rose@dkfz.de

the liver actually contributes in many aspects of metabolic control during fasting, and molecular mechanisms therein, remains largely unknown. It is known that the liver is the primary site of glucose production and can provide alternate substrates such as ketone bodies under fasting conditions (van den Berghe, 1991). Interestingly, the liver transcriptome starvation response correlates with lifespan prolonging processes (Bauer *et al*, 2004) and it is hypothesised that resistance to stress is an important determinant of survival and longevity (Calabrese *et al*, 2011). Hence, one strategy to uncover the molecular mechanisms contributing to metabolic dysfunction could be to dissect the liver transcriptome to uncover select genes which are aberrantly regulated during fasting. To this end, our studies here identify a member of the "growth arrest and DNA damage-inducible" (GADD45) gene family, namely GADD45β, as one such gene. Importantly, there are no studies that have systematically examined the role of GADD45β in metabolic function *in vivo*, and here, we demonstrate that liver GADD45β acutely regulates fatty acid handling under fasting stress, ultimately coordinating proper metabolic function under conditions of chronic nutrient oversupply.

## Results

### An altered lipid profile in mouse models of metabolic dysfunction is most pronounced in the fasted state

To examine the molecular bases of metabolic (dys)regulation during fasted and fed states, we required suitable mouse models. To this end, we examined mouse models of obesity (Kanasaki & Koya, 2011) including the obese/diabetic *db/db* mouse (Figs 1A–E and EV1A–D), pre-diabetic young New Zealand Obese mice (Figs 1F–N and EV1E–H) and aged mice (Figs 1K–O and EV1I–L). In particular, we subjected these mouse models to a fasting-refeeding regimen and could demonstrate heightened blood glucose (BG; Fig EV1A, E and I) most prominent in the fed and but also in the fasted state. In contrast, although there was evidence of higher serum non-esterified fatty acids (NEFA; Fig 1A, F and K) in the fed state, surprisingly these levels were lower in models of metabolic dysfunction in the fasted state. This pattern was also reflected in the level of triglycerides (TG; Fig 1B, G and L), ketone bodies (KB; Fig 1G, H and M) as well as medium-chain (MCAC; Fig 1D, I and N) and long-chain (LCAC; Fig 1E, J and O) acylcarnitines, with only mild differences in free- (Fig EV1B, F and J), acyl- (Fig EV1C, G and K) and short-chain (SCAC; Fig EV1D, H and L) acylcarnitines.

We then examined liver-specific metabolic control in a separate cohort using *ex vivo* liver metabolic tracing. In particular, long-chain fatty acid (LCFA) uptake was higher in the fasted state, and this was exacerbated in *db/db* mice (Fig 1P). The higher fasting LCFA uptake in WT mice was largely explained by higher LCFA oxidation (Fig 1Q). However, in the *db/db* mice, the heightened LCFA oxidation could not account for the higher uptake rate, meaning that non-oxidative LCFA disposal was enhanced. We examined incorporation into lipids but this was unchanged (data not shown). As fatty acids can indirectly affect hepatic glucose production (Ross *et al*, 1967), we examined liver glucose output, and indeed, it was higher in *db/db* mice, particularly in the presence of exogenous NEFA (Appendix Fig S1).

Taken together, our preliminary studies exemplify that systemic lipid metabolism is particularly disturbed in the fasted state in mice with metabolic dysfunction, with either enhanced clearance or reduced production of systemic NEFA and TG, the former of which might be explained by a liver-specific mechanism. Thus, our subsequent studies were focussed on the discovery of a molecular control point regulating (mal)adaptive lipid metabolism during fasting.

### Liver GADD45β is an inflexibly regulated gene transcript upon fasting stress in multiple mouse models of metabolic dysfunction

Previous studies of metabolic tissue transcriptomes in rodents (Bauer *et al*, 2004; Sokolovic *et al*, 2008; Hakvoort *et al*, 2011; Zhang *et al*, 2011; Schupp *et al*, 2013) identified members of the *Gadd45* gene family potently affected by fasting stress. Given that there are three members of the GADD45 family, we then decided to take a targeted approach to the discovery of novel molecular regulatory mechanisms, and examine the expression of all members in multiple models of metabolic dysfunction. To this end, we examined mRNA expression in the liver from the *db/db* mice (Fig 2A–C) and observed an upregulation of *Gadd45b* (~12-fold; Fig 2B) and *Gadd45g* (~sevenfold; Fig 2C), which was largely blunted in the liver of db/db mice; with only mild regulation of *Gadd45a* (Fig 2A). Furthermore, we employed less severe models of metabolic dysfunction such as a monogenic model of obesity-driven pre-diabetes (Kanasaki & Koya, 2011), the young *ob/ob* mouse (data not shown), a polygenic model of obesity-driven pre-diabetes, the young New Zealand Obese mouse (NZO; Fig 2D–F) and aged mice (Fig 2G–I). In these models, we could observe a consistent pattern of regulation of *Gadd45b* (Fig 2A, E and H), but not *Gadd45g* (Fig 2C, F and I), with again only mild regulation of *Gadd45a* (Fig 2A, D and G). Importantly, similar to the *Gadd45b* mRNA, which we could show

---

**Figure 1.  The dysregulated lipid metabolic phenotype of mouse models of metabolic dysfunction is most pronounced in the fasted state.**

A–O  Male 12-weeks-old wild-type (WT; C57Bl/6J) or obese/diabetic monogenic (*db/db*; BKS.Cg-m$^{+/+}$ Lepr DB/J; *n* = 4/group, A–E), New Zealand Black (NZB) and polygenic obese/pre-diabetic New Zealand Obese (NZO; *n* = 4/group, F–J), as well as young (i.e. 3 months) and aged (i.e. 22 months; *n* = 5/group, K–O), mice were fed *ad libitum* (fed) or fasted for 24 h (fasted). Serum non-esterified fatty acids (A, F, K), triglycerides (B, G, L) and ketone bodies (C, H, M) were measured. In addition, serum acylcarnitine profiling was conducted and medium-chain (D, I, N) and long-chain (E, J, O) acylcarnitine concentrations are shown.

P–S  In another cohort of mice, *ex vivo* long-chain fatty acid (LCFA) metabolism, including uptake (P), oxidation (Q) and non-oxidative LCFA disposal (NOFAD; R), in precision-cut liver slices from fed and fasted WT and db/db mice (*n* = 3/group; 4 slices per mouse), were determined. In addition, in slices from fasted mice, glucose output was determined in the presence of incubation with NEFA (BSA-NEFA) or vehicle (BSA) (S).

Data information: Data are mean ± SEM. Effect of genotype/age, *$P < 0.05$, **$P < 0.01$, ***$P < 0.001$. Effect of nutritional state: #$P < 0.05$, ##$P < 0.01$, ###$P < 0.001$. The statistical test used and respective *P*-value outputs can be found in Appendix Table S1.

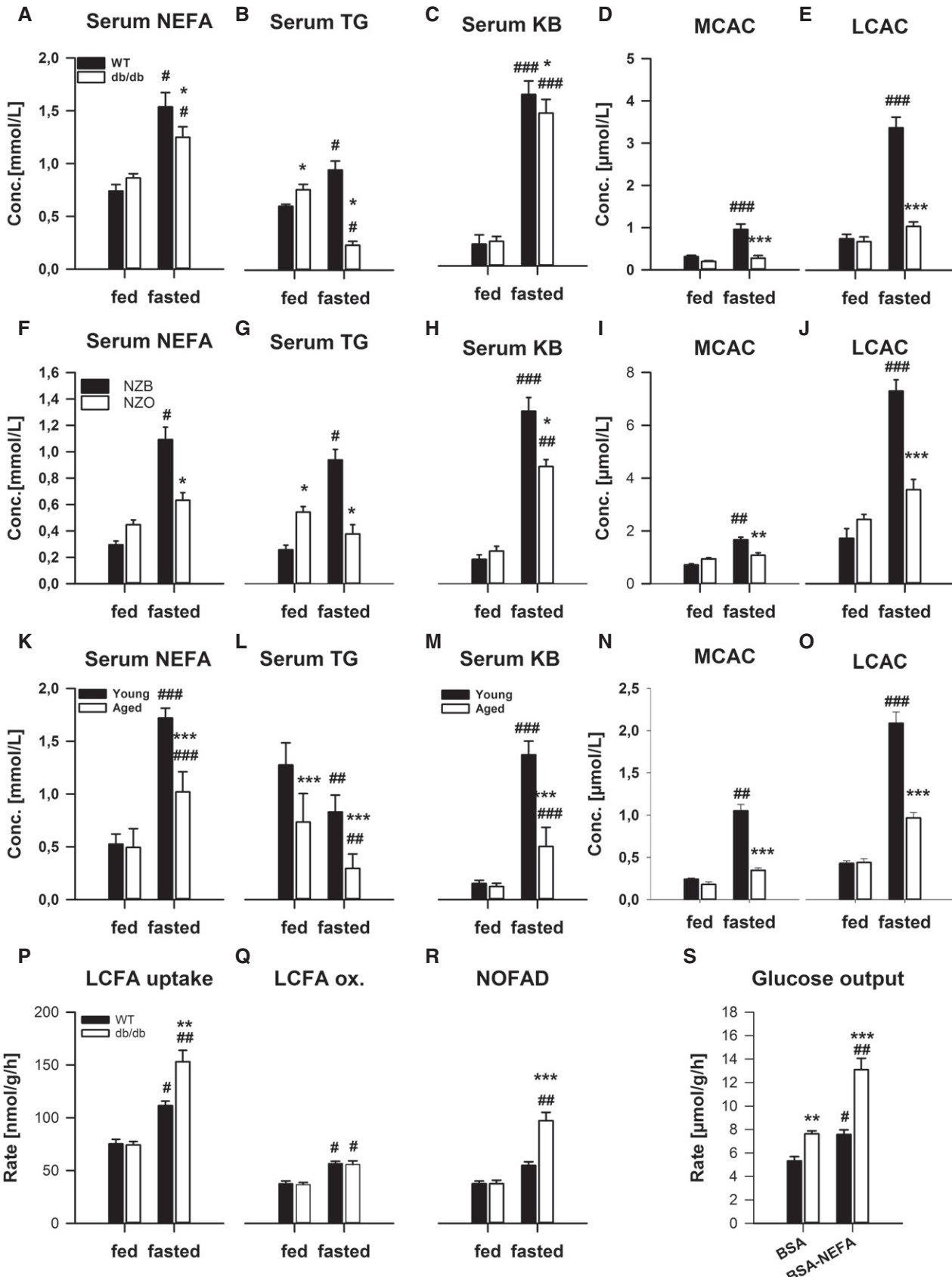

Figure 1.

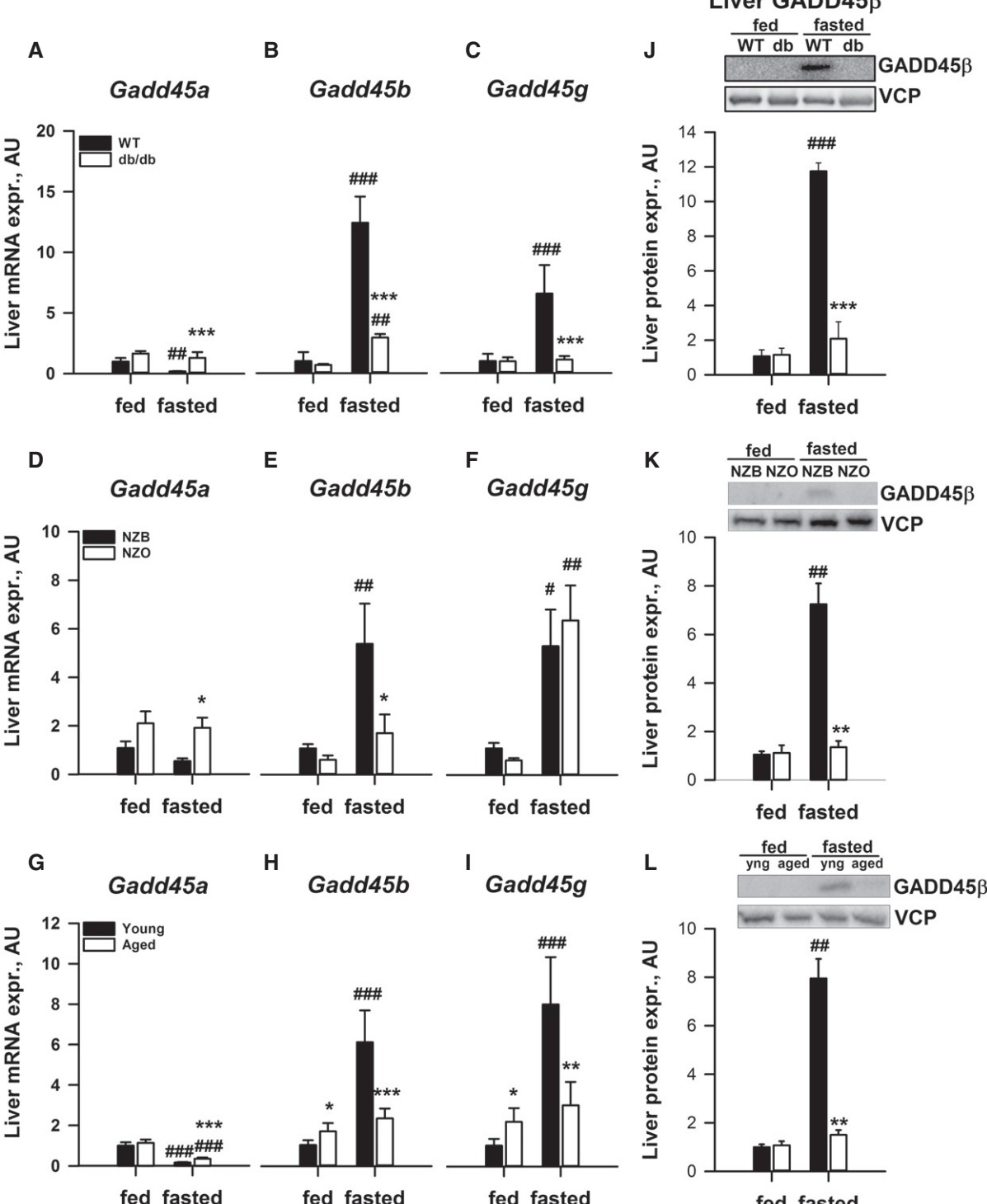

**Figure 2.  Liver/hepatocyte *Gadd45b* expression is consistently dysregulated upon fasting in several mouse models of metabolic dysfunction.**

A–I   Liver mRNA expression of growth arrest and DNA damage-inducible 45 alpha (*Gadd45a*; A, D, G), beta (*Gadd45b*; B, E, H) and gamma (*Gadd45g*; C, F, I) was measured in fed and fasted obese/diabetic monogenic (*db/db* (n = 4/group); A–C), obese/pre-diabetic polygenic New Zealand Obese (NZO (n = 4/group; D–F) and aged C57Bl/6J (22 months; n = 5/group; G–I) as well as corresponding lean, young wild-type (WT) mice (n = 4/group). Matched controls for the NZO and aged mice were New Zealand Black (NZB) and 12-weeks-old C57Bl/6J mice, respectively.

J–L   Protein expression of liver GADD45β, as well as the housekeeping protein valosin-containing protein (VCP), was measured from liver samples. Number of replicates for each sample set are outlined above.

Data information: Data are mean ± SEM. Effect of genotype, *P < 0.05, **P < 0.01, ***P < 0.001. Effect of nutritional state: #P < 0.05, ##P < 0.01, ###P < 0.001. The statistical test used and respective P-value outputs can be found in Appendix Table S1.

is dynamically regulated upon fasting and refeeding (Appendix Fig S1A), we could observe a similar pattern of upregulation of GADD45β protein in the liver of fasted mice, which was absent in the liver of models of metabolic dysfunction (Fig 2J–L). In addition, although we could show mild differential regulation in perigonadal white adipose tissue (pgWAT; Appendix Fig S1B–D), brown adipose tissue (BAT; Appendix Fig S1E–G) and gastrocnemius complex skeletal muscle (GCM; Appendix Fig S1H–J) of *Gadd45a* (Appendix Fig S1B, E and H), *Gadd45b* (Appendix Fig S1C, F and I) and *Gadd45g* (Appendix Fig S1D, G and J) between fed and fasted states comparing lean and obese/T2D mice, none of these regulations were as striking as observed with the liver *Gadd45b* (Fig 2).

### GADD45β affects metabolic regulation under conditions of heightened lipid metabolism

Given that we could observe a stark regulation of liver GADD45β under fasting stress, we next tested whether GADD45β expression/activity affects the ability of the organism to handle such nutritional stress by subjecting whole-body GADD45β knockout mice to starvation and refeeding. In particular, absolute levels as well as changes in body mass (Fig EV2A), food intake (Fig EV2B) and physical activity (Fig EV2C) were unaffected in GADD45β KO mice. Furthermore, systemic oxidative metabolism, as reflected at the rate of $O_2$ consumption (Fig 3A), $CO_2$ production (Fig 3B) and respiratory exchange ratio (Fig 3C), was also unaffected. In a separate study, we could demonstrate that there was no compensatory upregulation of the other GADD45 family members with loss of GADD45β (Fig EV2D) and could confirm the lack of effect of GADD45β loss on body mass regulation upon starvation (Fig EV2E). When we examined the masses of selected metabolic organs/tissues, there were also no differences (Fig EV2F). On the other hand, while we could not observe any differences in selected liver metabolites (Fig EV3G) or blood serum levels of glucose (Fig EV2H), TG (Fig 3D), ketone bodies (Fig 3E), glycerol (Fig EV2I) and acylcarnitines (Fig EV2J), we could observe a significantly lower increase in serum NEFA (Fig 3F) with a much higher level of accumulation of TG (~twofold; Fig 3G) in the liver of GADD45β KO mice, pointing to a role for GADD45β in coordinating an aspect of systemic lipid metabolism under starvation stress. Importantly, even though we could observe changes in *Gadd45γ* upon starvation (Fig 2), there was no discernible differential metabolic phenotype in starved GADD45γ KO mice (Appendix Fig S2).

Given that we could detect changes in lipid metabolism during starvation, we next directly tested lipid homoeostasis by conducting an oral lipid tolerance test. Strikingly, systemic lipid clearance was accelerated in fasted GADD45β KO mice, with substantially lower serum TG (Fig 3H) and NEFA (Fig 3I). Perhaps surprisingly, blood glucose levels rose to a greater extent in the GADD45β KO mice following oral lipid provision (Fig 3J), indicating that GADD45β might be a molecular regulator of lipid–glucose metabolic crosstalk. To test this further, we conducted a chronic high-fat diet (HFD) study, which should then reveal whether GADD45β is operative in affecting metabolic control during more mild but chronic fasting-feeding rhythms, and while there were no substantial effects of GADD45β loss on glucose homoeostasis on the normal-fat diet (NFD), there was impaired glucose homoeostasis upon high-fat diet treatment as demonstrated by intraperitoneal insulin tolerance testing (ipITT; Fig 3K) as well as fasting glucose (BG; Fig 3J), insulin (Fig 3M) and a calculated insulin resistance index (HOMA-IR; Fig 3N), which were independent of compensatory GADD45 expression, body and liver mass in the *ad libitum* fed state (Appendix Fig S3).

### Liver GADD45β modulates metabolic regulation under conditions of heightened lipid metabolism

We next wanted to test whether liver-specific GADD45β affects systemic metabolism. Indeed, we could demonstrate that similar to the lipid tolerance test results (Fig 3C and D), *ex vivo* liver lipid handling is affected by GADD45β loss (Fig 4A–D), selectively in the fasted state when *Gadd45b* is upregulated (Fig 2). In particular, long-chain fatty acid uptake was enhanced (LCFA; Fig 4A), without effects on LCFA oxidation (Fig 4B), and thus largely driven by an enhanced non-oxidative metabolism (NOFAD; Fig 4C). Similar to the exaggerated glycemic response to oral lipid (Fig 3E), the glucose production rate from liver *ex vivo* was exacerbated by GADD45β loss (Fig 4D), highlighting a potential role of GADD45β in coordinating proper liver fatty acid–glucose metabolism crosstalk.

To examine whether the upregulation of *Gadd45b* mRNA during fasting was occurring within the parenchymal cell of the liver, namely the hepatocyte, we performed a liver cell-type fractionation experiment from livers of fasted and fed mice. While the fractionation of hepatocytes leaked into the non-parenchymal fraction to a mild extent (*Alb*; Fig EV3A), the hepatocyte fraction was devoid of non-parenchymal markers of endothelial (*Cd31*; Fig EV3B), stellate (*Des*; Fig EV3C) and Kupffer (*Emr1*; Fig EV3D) cells, and there was only an upregulation of *Gadd45b* in the hepatocyte fraction (Fig 4E), indicating that the increase of *Gadd45b* expression within the liver upon fasting occurs within the hepatocyte.

Given the above findings, we then wanted to assess whether GADD45β affects metabolic control in a liver-specific manner *in vivo*. As such, we conducted a study whereby we silenced

**Figure 3. Systemic GADD45β deletion affects metabolic regulation under conditions of heightened lipid metabolism.**

A–G  Male GADD45β$^{+/+}$ (WT; n = 6) or GADD45β$^{-/-}$ (KO; n = 4) mice were fed *ad libitum* (fed) or fasted for 24 h (fasted) and subsequently refed for 24 h. $O_2$ consumption rate (A), CO2 production rate (B) and respiratory exchange ratio (C) were measured by indirect calorimetry. In a distinct cohort, male GADD45β$^{+/+}$ (WT; n = 5) or GADD45β$^{-/-}$ (KO; n = 8) were fed *ad libitum* (fed) or fasted for 24 h (fasted). Serum non-esterified fatty acids (D) and ketone bodies (E) as well as serum (F) and liver (G) triglycerides (TG) were measured.

H–J  Serum TG (H), NEFA (I) and blood glucose (BG, J) concentrations during an oral lipid tolerance test in overnight fasted GADD45β$^{+/+}$ (WT; n = 5) or GADD45β$^{-/-}$ (KO; n = 5) mice.

K–N  Blood glucose excursion during and intraperitoneal insulin tolerance test (K) as well as fasting blood glucose (L), serum insulin (M) and HOMA-IR (N) in GADD45β$^{+/+}$ (WT; n = 6) or GADD45β$^{-/-}$ (KO, n = 9) chronically fed a normal- (NFD) or high (HFD)-fat diet.

Data information: Data are mean ± SEM. Effect of genotype, *$P < 0.05$, **$P < 0.01$, ***$P < 0.001$. Effect of nutritional state: #$P < 0.05$, ##$P < 0.01$, ###$P < 0.001$. The statistical test used and respective $P$-value outputs can be found in Appendix Table S1.

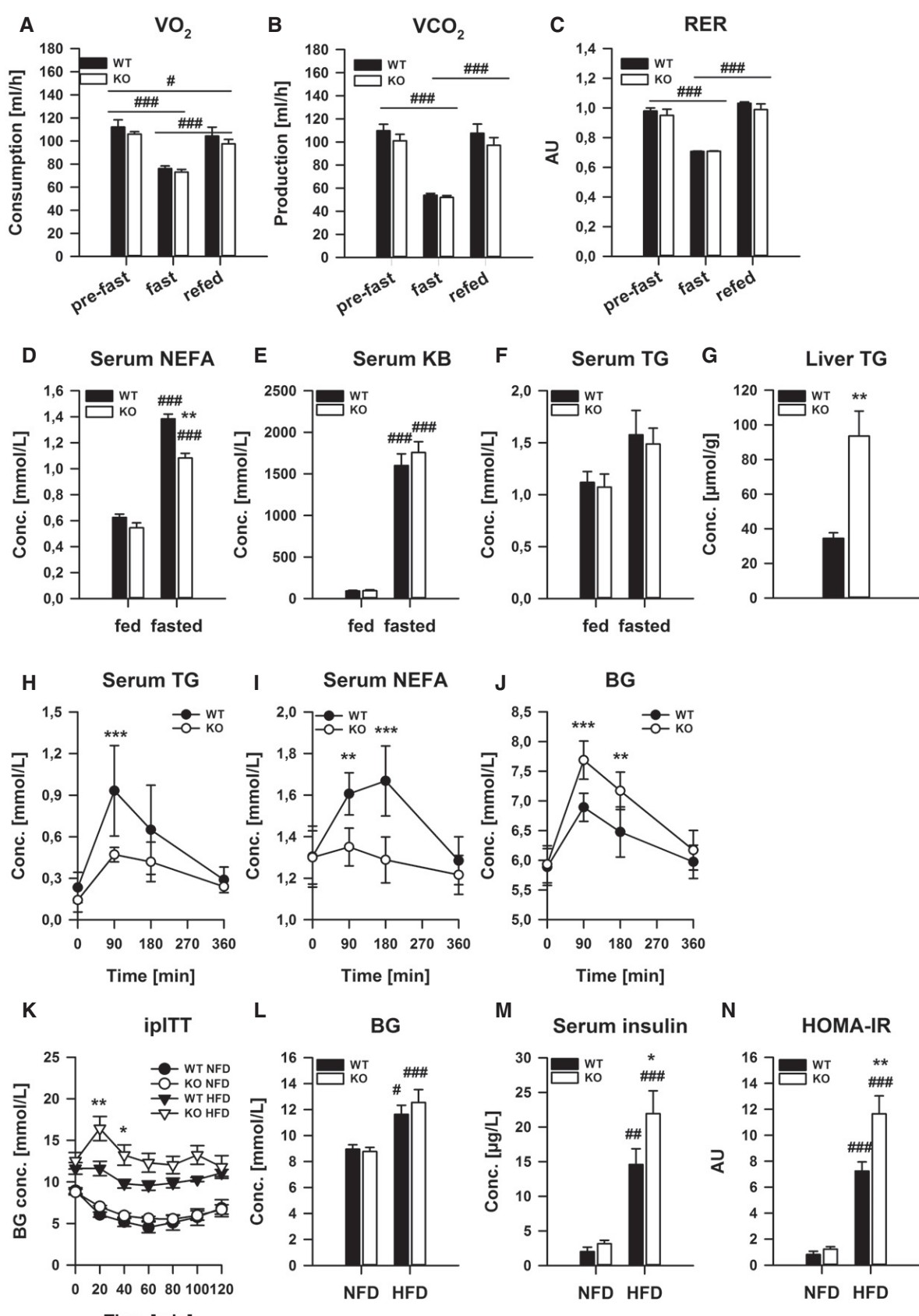

**Figure 3.**

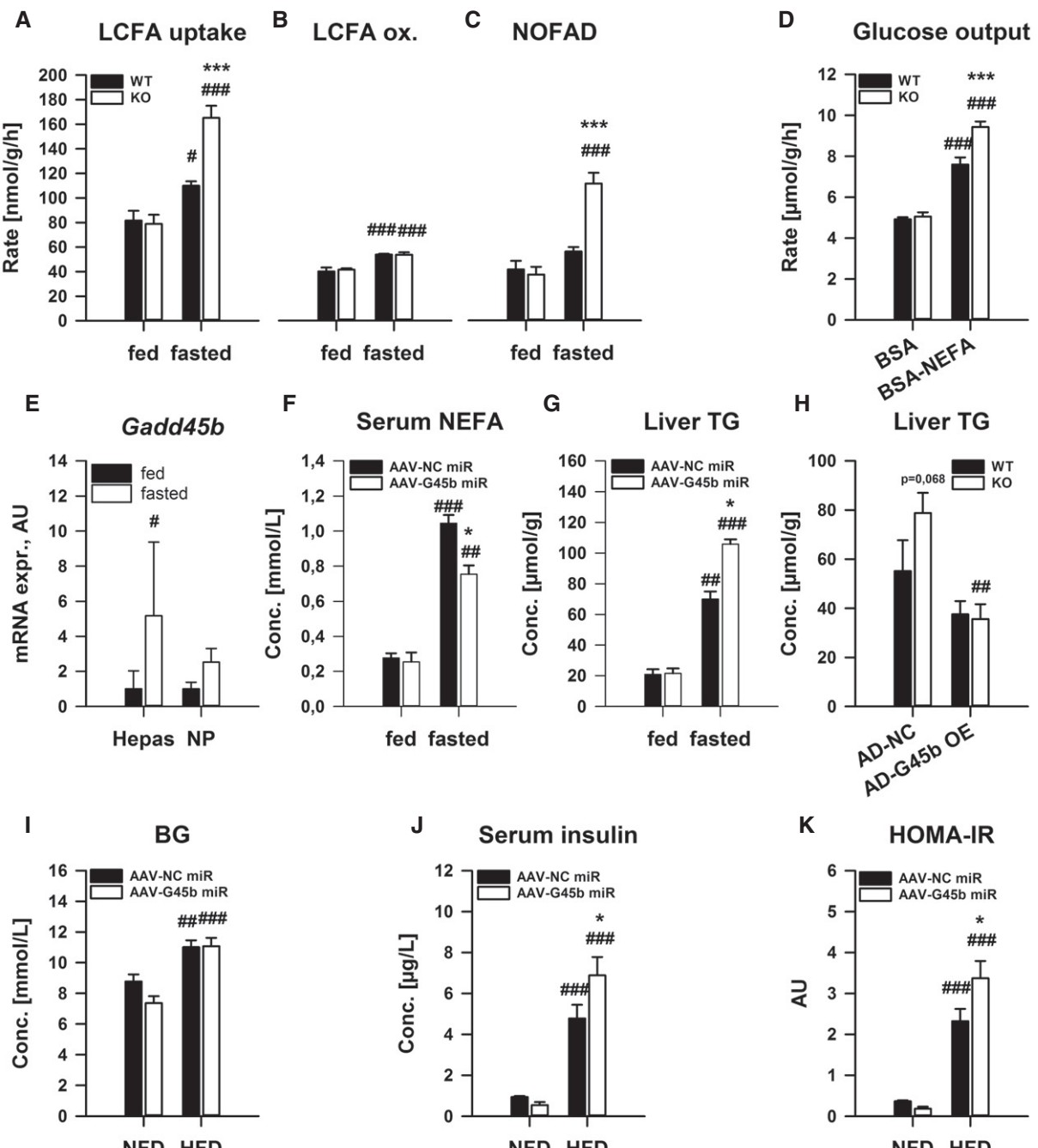

**Figure 4.  Liver GADD45β modulates metabolic regulation under conditions of heightened lipid metabolism.**

A–D    Male, GADD45β$^{+/+}$ (WT) or GADD45β$^{-/-}$ (KO) mice were fed *ad libitum* (fed) or fasted for 24 h (fasted), and *ex vivo* liver slice long-chain fatty acid (LCFA) metabolism was measured including LCFA uptake (A), oxidation (B) and non-oxidative LCFA disposal (NOFAD) was calculated (C). In addition, glucose production was measured (D) in the presence (BSA-NEFA) or absence (BSA) of extracellular fatty acids (A–D: n = 4/group with four liver slices per mouse liver).

E    *Gadd45b* mRNA expression was measured from fractionated parenchymal hepatocytes (Hepas) as well as non-parenchymal cells (NP) from C57Bl/6J mice fed *ad libitum* (fed) or fasted for 24 h (fasted); n = 3/group.

F, G    Male C57Bl/6J mice with (AAV-G45b miR) or without (AAV-NC miR) liver/hepatocyte-restricted GADD45β silencing were fed or fasted for 24 h (n = 6/group), and serum levels of non-esterified fatty acids (NEFA; F) as well as liver triglyceride (TG) concentration (G) were measured.

H    Male GADD45β$^{+/+}$ (WT; n = 16) or GADD45β$^{-/-}$ (KO; n = 15) mice fasted for 24 h (fasted) with (AD-G45b OE) or without (AD-NC) liver-restricted GADD45β overexpression (n = 7–8/group). Liver TG concentration was measured.

I–K    Male C57Bl/6J mice with (AAV-G45b miR; n = 15) or without (AAV-NC miR, n = 13) liver/hepatocyte-restricted GADD45β silencing were chronically fed a normal-fat diet (NFD) or high-fat diet (HFD) (n = 6–8/group). Fasting blood glucose (I) and serum insulin (J) were measured from which HOMA-IR was calculated (K).

Data information: Data are mean ± SEM. Effect of genotype/viral manipulation, *P < 0.05, **P < 0.01, ***P < 0.001. Effect of nutritional state: $^{#}$P < 0.05, $^{##}$P < 0.01, $^{###}$P < 0.001. The statistical test used and respective P-value outputs can be found in Appendix Table S1.

*Gadd45b* in a liver/hepatocyte selective manner (Graham *et al*, 2008; Rose *et al*, 2011) via AAV-mediated delivery of a *Gadd45b*-specific miRNA. Using this strategy, we were able to substantially blunt the expression of *Gadd45b*, particularly during fasting (Fig EV3E). Similar to the results from the germline whole-body KO (Fig 3), there was a blunting of the higher serum NEFA (Fig 4F) and exacerbation of the higher liver TG (Fig 4G) in the fasted liver-specific *Gadd45b* silenced mice. To confirm that hepatic GADD45β affects lipid metabolism, we conducted a study whereby we overexpressed GADD45β in the liver in GADD45β KO mice (Fig EV3F). Indeed, GADD45β re-introduction into the liver of whole-body GADD45β KO, despite not affecting serum NEFA (Fig EV3G), affected serum TG (Fig EV3H) and alleviated the heightened accumulation of TG in the liver of GADD45β KO mice upon fasting (Fig 4H). To test this further, we conducted a chronic high-fat diet (HFD) study, which should then reveal whether GADD45β is operative in affecting metabolic control during more mild but chronic fasting-feeding rhythms. While there were no substantial effects of GADD45β loss on biometric parameters (Fig EV3I–K), similar to the whole-body KO study (Fig 3), hepatocyte-specific GADD45β silencing worsened the progression of insulin resistance upon high-fat diet-induced obesity (Fig 4I–K).

### Impaired liver GADD45β expression correlates with metabolic dysfunction in obesity-driven type 2 diabetes in mouse and man

Prompted by our results that GADD45β loss can affect glucose homoeostasis in obesity (Figs 3 and 4), we tested whether restoration of liver GADD45β can improve metabolic homoeostasis in type 2 diabetes. To this end, we employed our prior strategy and overexpressed GADD45β in the liver of obese/diabetic *db/db* mice (Fig EV4A). While biometrics (Fig EV4B–D) were not drastically affected, fasting blood glucose (Fig 5A) and insulin (Fig 5B) were lower in diabetic mice, which led to an overall reduction in systemic insulin resistance (HOMA-IR; Fig 5C). Since GADD45β expression in liver affects lipid and glucose homoeostasis in mice, we wondered if the same were true in humans. Indeed, during fasting, *GADD45B* expression was significantly lower in livers of type-2 diabetic patients (T2D) when compared with aged-matched individuals with normal glucose tolerance (NGT; Fig 5D). Furthermore, although there was a non-significant negative correlation with HOMA-IR (Fig EV4E), liver *GADD45B* expression negatively correlated with fasting TG (Fig 5E) and glucose (FPG; Fig 5F) levels. Hence, similar to mice, effective *GADD45B* expression during fasting may also confer proper metabolic control in humans.

### Liver GADD45β controls liver fatty acid handling by cytosolic FABP1 retention

Given the effects of GADD45β on metabolism, we then searched for a mechanism by which it might exert such control. Given that prior studies have demonstrated a role for GADD45β to control various aspects of cellular functions though transcriptional or signalling mechanisms we first focussed on this. In particular, overexpression of GADD45β in the livers of diabetic mice did not enhance key insulin signalling nodes (Fig EV5A and B). Furthermore, although previously implicated in other studies (Keil *et al*, 2013; Tian & Locker, 2013), GADD45β expression did not affect autophagy,

mTORC1, MAPK or ER stress signalling pathways (Fig EV5C and D). Nor did GADD45β (Fig 6A) affect expression of key genes involved in liver fatty acid transport/metabolism at the mRNA (Fig 6B) or protein (Fig EV5C and D) level. Consistent with a role for localisation of key FA transport/handling proteins in metabolic control (Glatz *et al*, 2002; Kazantzis & Stahl, 2012), we could, however, observe a role for GADD45β in regulating FABP1, but not FATP2 nor CD36, localisation (Fig 6C). In particular, FABP1 localisation was redistributed from the cytoplasm to the low-density microsomal (i.e. plasma membrane and microsomes) membranes with a lack of GADD45β (Figs 6C and D, and EV5E), which could be reversed by re-expression of GADD45β in a liver-specific manner (Fig 6E). Furthermore, similar to the liver of GADD45β knockout mice, obese/diabetic db/db mice had a redistribution of FABP1 towards microsomes and away from the cytosol, which could be reversed by GADD45B overexpression (Fig 6F).

Given that immunoprecipitation experiments demonstrated that GADD45β can physically bind in a complex with FABP1 (Fig 6G) and that GADD45β is exclusively expressed in the cytoplasm (Fig 6C), GADD45β might operate as a cytosolic adapter molecule for FABP1. Lastly, expression of GADD45β and associated altered localisation of FABP1 consistently negatively correlated with liver levels of activated long-chain fatty acids in multiple experiments (Fig 6H–J), consistent with a direct role for GADD45β to modulate hepatocellular FA metabolism.

## Discussion

Here, we demonstrate that liver GADD45β expression is dynamically regulated upon fasting stress, perhaps by increased oxidative stress (Zhang *et al*, 2013; Kim *et al*, 2014), where it aids in the coordination of liver lipid metabolism, by limiting hepatocellular fatty acid (FA) uptake via cytosolic FABP1 retention. Furthermore, in mouse and human metabolic dysfunction such as type 2 diabetes, obesity/pre-diabetes and the aged, liver GADD45β expression is dysregulated thereby contributing to aberrant lipid/glucose homoeostasis.

The GADD45 gene family composes of three structurally and functionally related genes named as GADD45α, GADD45β and GADD45γ, encoding small, highly acidic, nuclear proteins (Liebermann & Hoffman, 2002). Depending on the cellular stress and the physical interactions with other proteins, GADD45 proteins serve as stress sensors, which participate in cell cycle arrest, apoptosis and cell survival (Liebermann & Hoffman, 2008). Despite their structural and physical similarities, their response to stress conditions is cell type and stimulus specific (Yang *et al*, 2009). Indeed, similar to prior studies (Bauer *et al*, 2004; Zhang *et al*, 2011), here, we could show that fasting preferentially promotes GADD45β expression in the liver, whereas others have demonstrated that physical exertion (Hoene & Weigert, 2010), cold stress (Gantner *et al*, 2014) and atrophy (Ebert *et al*, 2012) activated GADD45 isoform expression in other diverse metabolic tissues, with functional metabolic effects. Furthermore, there are several lines of evidence that GADD45β is involved in hyperplasia of hepatocytes in liver regeneration (Columbano *et al*, 2005; Papa *et al*, 2008; Tian *et al*, 2011), and although hepatic TG accumulation occurs during liver regeneration, the effects GADD45β on lipid metabolism identified here are unlikely to

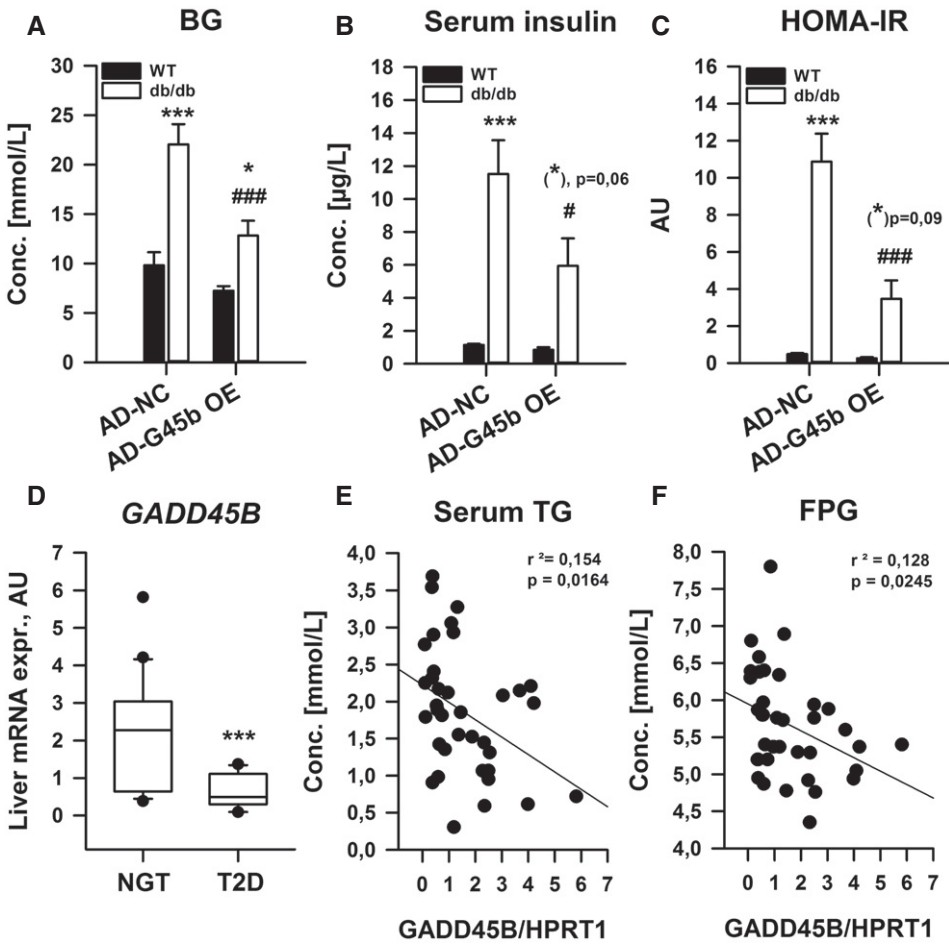

**Figure 5. Liver GADD45β expression modulates glucose homoeostatic control in type 2 diabetes.**

A–C  Male 12-weeks-old wild-type (WT; C57Bl/6J) or obese/diabetic (db/db; BKS.Cg-m$^{+/+}$ Lepr DB/J) mice with (AD-G45b OE) or without (AD-NC) prior liver-restricted GADD45β over-expression were fasted, and blood glucose (A) and serum insulin (B) were measured from which HOMA-IR was calculated (C) (n = 4–6/group). Data are mean ± SEM. Effect of genotype/viral manipulation, *P < 0.05, **P < 0.01, ***P < 0.001. Effect of nutritional state: #P < 0.05, ##P < 0.01, ###P < 0.001.

D–F  Liver GADD45B mRNA expression from men with (T2D) or without (NGT) type 2 diabetes (n = 14–23/group). ***P < 0.001. Scatter plots fasting plasma triglycerides (E) and glucose (F) in correlation with liver GADD45B mRNA expression (n = 37). Inserts show $r^2$ values and P-values from Spearman's correlation test.

Data information: The statistical test used and respective P-value outputs can be found in Appendix Table S1.

be of relevance in this setting (Newberry et al, 2008). Even though GADD45β has been linked to epigenetic control (Ma et al 2009), it is unlikely that the altered metabolism observed within GADD45β knockout mice results from developmental "programming" as we could show congruent phenotypes when GADD45β was manipulated in adult mice.

Through liver-specific GADD45β manipulation in vivo, we demonstrated that the altered liver fatty acid metabolism in obesity/diabetes and ageing might result from altered liver GADD45β expression, particularly during the fasted state. In particular, fasted obese/T2D mice had enhanced liver LCFA uptake, with a similar phenotype observed in fasted GADD45β KO mice, suggesting that the blunted regulation of liver GADD45β and metabolic derangements in mouse models of metabolic dysfunction may be causally linked. We speculate that the persistent fast-feeding cycles intermittently activate liver GADD45β expression, leading to proper coordination of liver lipid metabolism during fasting, ultimately impacting the maintenance of metabolic health, particularly when dietary lipid/nutrient intake is high. Indeed, liver lipid metabolism is intimately linked to systemic glucose control (Perry et al, 2015).

While it is conceivable that GADD45β could achieve its effects via both cell-autonomous and non-autonomous mechanisms, we could demonstrate that GADD45β modulates LCFA metabolism ex vivo, indicating that the former is likely to be the case, and that factors within hepatocytes per se are likely to be affected by GADD45β. In particular, unlike other studies that have shown a role for the activation of transcriptional co-activator PGC1α to retard metabolic dysfunction by promoting fatty acid oxidation (Morris et al, 2012), we could demonstrate that fasting liver LCFA oxidation was not affected, but rather non-oxidative FA metabolism independent of storage was involved in GADD45β-mediated effects on hepatic lipid metabolism. This is reminiscent of studies of skeletal muscle, where an unknown FA metabolic fate correlates with obesity-related metabolic dysfunction (Koves et al, 2008), and probably relates to a relative mitochondrial substrate "overload" (Muoio & Neufer, 2012), which seems to be also present in the liver (Satapati et al, 2012).

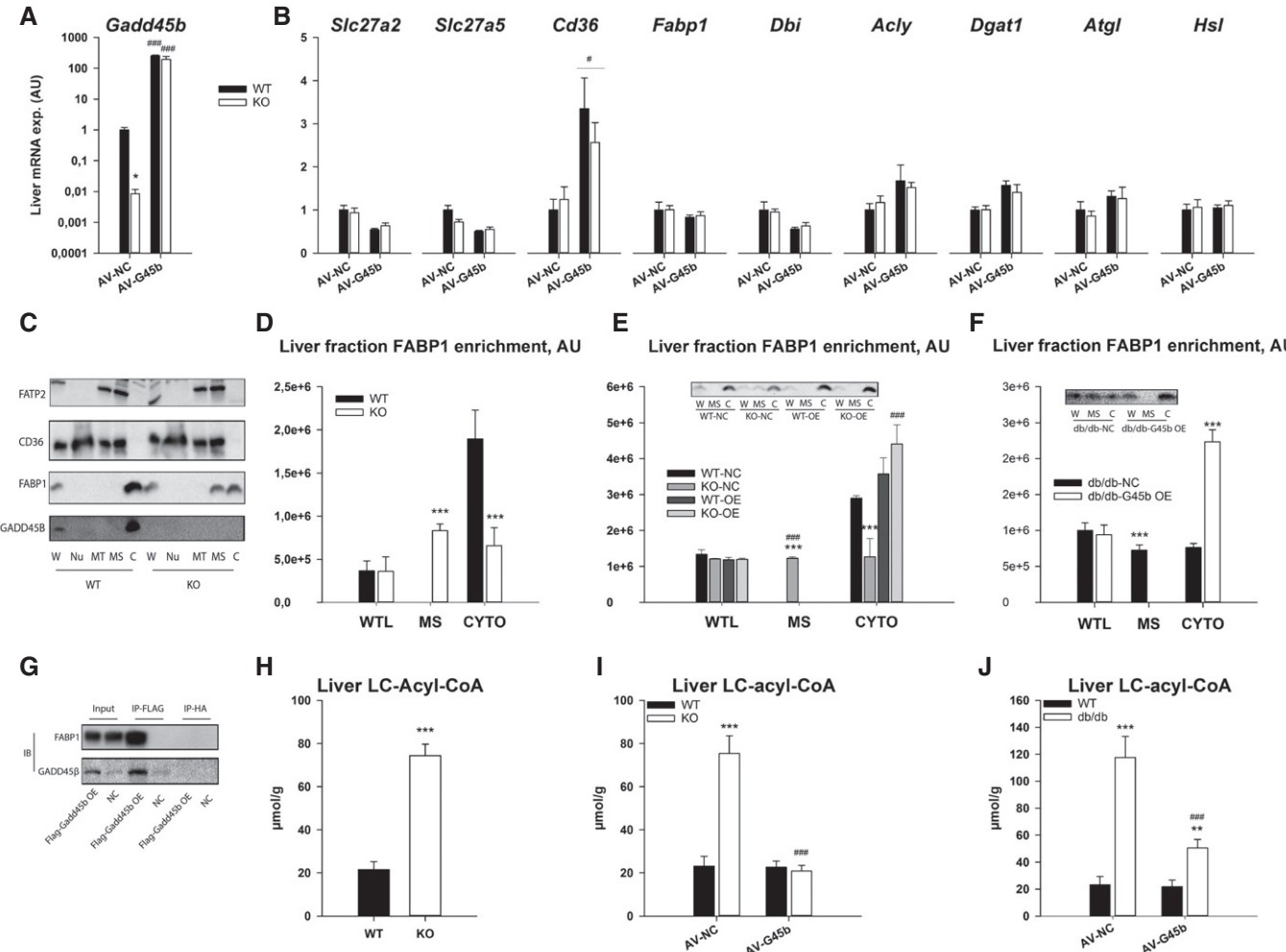

**Figure 6.  Liver GADD45β controls liver fatty acid handling by cytosolic FABP1 retention.**

A, B    Male GADD45β$^{+/+}$ (WT; $n$ = 16) or GADD45β$^{-/-}$ (KO; $n$ = 15) mice fasted for 24 h (fasted) with (AV-G45b OE) or without (AV-NC) liver-restricted GADD45β over-expression ($n$ = 7–8/group). Liver mRNA expression of Gadd45b (A) as well as fatty acid metabolic genes (B) encompassing transport (Slc28a2, Slc27a5, Cd36), intracellular binding (Fabp1, Dbi) and metabolism (Acly, Dgat1, Atgl, Hsl).

C    Representative immunoblots of FATP2, CD36, FABP1 and GADD45β from liver whole tissue lysate (W) as well as fractionated organelles/intracellular structures including nuclei (N), mitochondria (MT), microsomes (MS) and cytoplasm (C), from GADD45β$^{+/+}$ (WT) and GADD45β$^{-/-}$ (KO) mice.

D    Quantified band densities of FABP1 enrichment from fractions in C ($n$ = 4/group).

E    Liver fraction enrichment of FABP1 from male GADD45β$^{+/+}$ (WT) or GADD45β$^{-/-}$ (KO) mice fasted for 24 h with (AD-G45b OE) or without (AD-NC) liver-restricted GADD45β over-expression ($n$ = 4/group). Insert shows a representative FABP1 immunoblot.

F    Liver fraction enrichment of FABP1 from obese/diabetic male *db/db* mice fasted for 24 h with (AD-G45b OE) or without (AD-NC) liver-restricted GADD45β over-expression ($n$ = 4/group). Insert shows a representative FABP1 immunoblot.

G    FABP1 and GADD45B immunoblots from Flag immunoprecipitations (IP-FLAG) or mock IP (IP-HA) from liver input samples from mice with (AD-G45b OE) or without (AD-NC) liver-restricted GADD45β over-expression. Shown is a representative immunoblot from 3 separate experiments using 3 different input samples per condition.

H–J    Liver tissue long-chain acyl-CoA (LC-acyl-CoA) concentrations were determined in GADD45β$^{+/+}$ (WT) or GADD45β$^{-/-}$ (KO) mice (H; $n$ = 6/group) with (AD-G45b OE) or without (AD-NC) liver-restricted GADD45β over-expression (I; $n$ = 5/group). Liver LC-acyl-CoA concentrations were determined in wild-type (WT; C57Bl/6J) or obese/diabetic (*db/db*; BKS.Cg-m$^{+/+}$ Lepr DB/J) mice with (AD-G45b OE) or without (AD-NC) liver-restricted GADD45β over-expression (J; $n$ = 4/group).

Data information: Data are mean ± SEM. Effect of genotype, *$P$ < 0.05, **$P$ < 0.01, ***$P$ < 0.001. Effect of viral manipulation: #$P$ < 0.05, ##$P$ < 0.01, ###$P$ < 0.001. The statistical test used and respective $P$-value outputs can be found in Appendix Table S1.

Furthermore, chronically enhanced liver FA supply/uptake can nega-tively affect systemic metabolic function (Mittelman *et al*, 1997; Koonen *et al*, 2007; Doege *et al*, 2008; Falcon *et al*, 2010; Perry *et al*, 2015) the mechanisms by which are not fully resolved. On this, we could demonstrate that induced GADD45β restrained liver fatty acid uptake during fasting ultimately preventing an accumulation of

long-chain (LC) acyl-CoAs in the liver. Given that liver LC-acyl-CoAs levels positively correlate with insulin resistance (present work and (Chen *et al*, 1992; Kamath *et al*, 2011), they may be directly or indi-rectly related to aberrant glucose metabolism under conditions of enhanced fatty acid supply such as obesity (Li *et al*, 2010), the mech-anisms by which are likely to be diverse (Faergeman & Knudsen,

1997; Cooney *et al*, 2002; Li *et al*, 2010). Indeed, here we observed that restoration of low liver GADD45β expression in obese/diabetic mice reduced the higher liver LC-acyl-CoA amount in liver, which correlated with improved glucose homoeostasis.

Here, we demonstrate that GADD45β binds to and restrains FABP1 to the hepatocyte cytoplasmic compartment, ultimately preventing an accumulation of activated long-chain fatty acids. Importantly, although normally heavily abundant in the cytoplasm, redistribution of FABP1 has been described previously in other contexts (Wolfrum *et al*, 2001; Antonenkov *et al*, 2006). Consistent with a key role for FABP1 localisation in explaining the altered FA metabolism modulated by GADD45β expression, FABP1 is a highly abundant protein in the liver which has a role in cellular uptake of FAs and affects the development of insulin resistance in obesity (Atshaves *et al*, 2010). In particular, *in vitro* gain- and loss-of-function studies (Wolfrum *et al*, 2001; Linden *et al*, 2002; Newberry *et al*, 2003) have implicated FABP1 in hepatocellular FA uptake, and studies of germline FABP1 knockout mice demonstrate reduced hepatic FA uptake particularly during conditions of heightened FA supply such as fasting (Martin *et al*, 2003; Newberry *et al*, 2003), which is consistent with our studies here. Thus, we hypothesise that aberrant localisation of FABP1 towards microsomal membranes, including the surface membranes and microsomes, is a molecular event linking excess hepatic fatty acid uptake and low GADD45β expression during fasting.

Of note, the aberrant lipid metabolic phenotype, including serum NEFA, triglycerides, ketone bodies, acylcarnitines and liver triglycerides, of obese/aged mice was particularly observed in the fasted state, which corresponds to studies in humans demonstrating that differential metabolic phenotypes are best revealed by challenges (Krug *et al*, 2012). In flies, D-GADD45 overexpression results in increased longevity, and a hallmark of long-lived animals is resistance to stressors (Calabrese *et al*, 2011). Indeed, we (data not shown) and others (Kim *et al*, 2015) have demonstrated that GADD45β protects against liver damage upon toxicity stress and fasting activates pleiotropic adaptive processes that are beneficial to organismal health (Longo & Mattson, 2014). Given that liver toxicity response is exacerbated in obese/diabetic mice (Aubert *et al*, 2012), of which have dysregulated GADD45β expression, the coordination of metabolism by GADD45β may represent of molecular metabolic link in adaptive stress biology. Taken together, we hypothesise that GADD45β might represent a "vitagene" conferring hormetic metabolic adaptive processes (Calabrese *et al*, 2011), and that the concept of "metabolic inflexibility" (Storlien *et al*, 2004) may simply reflect a lack of response of such hormetic mechanisms (Kolb & Eizirik, 2012), of which liver GADD45β induction may be one component.

In summary, here, we identify a role of a liver transcript, GADD45β, in modulating systemic and liver-specific adaptive metabolism under nutrient-starvation stress, which ultimately aids in the coordination of metabolic homoeostasis upon chronic nutrient overload.

## Materials and Methods

### Mouse experiments

Mouse strains used included male wild-type C57Bl/6J (000664; Charles River Laboratories, DEU), *db/db* (12 weeks; 000642, BKS.Cg-m$^{+/+}$ Lepr DB/J, Charles River Laboratories, DEU), *ob/ob* (7 weeks; 000632, B6.Cg-*Lep*$^{ob}$/J, Jackson Laboratories, USA), New Zealand Black (7 weeks; NZB/BlNJ, 000993, Jackson Laboratories, USA) and New Zealand Obese (7 weeks; NZO/HlLtJ, 002105, Jackson Laboratories, USA). Furthermore, germline male GADD45β (B6.CgGADD45β$^{tm1Daa}$; Gupta *et al*, 2005) and female (XX) GADD45γ (B6.CgGADD45γ$^{tm1Mhol}$; Cai *et al*, 2006) knockout mice from $^{-/+} \times ^{-/+}$ crossings were used. Importantly, SNP marker testing demonstrated that the GADD45β mouse line was of 99.1% pure C57Bl/6 background strain (Charles River Genetic Testing Services; data not shown). The animals were housed according to international standard conditions with a 12-h dark–light cycle and regular unrestricted diet with free access to water if not stated otherwise. For studies with overexpression of GADD45β in the murine liver, $1 \times 10^9$ infectious units per recombinant adenovirus (AD) were administered via tail vein injection into either male *db/db* and C57Bl/6J control mice or GADD45β WT and KO littermates 1 week after acclimation on control diet, all at the age of 12 weeks. Another week later, mice were fasted for 24 h before sacrifice. In other studies, hepatocyte-specific GADD45β knockdown was accompanied with fasting or HFD treatments. After 1 week of acclimation on control diet, at the age of 9 weeks, $2 \times 10^{11}$ virus particles per adeno-associated virus (AAV) were administered via tail vein injection into male C57Bl/6J mice. The fasting study was carried out, half the mice from each virus group either fed *ad libitum* or were subjected for fasting for 24 h. For the HFD study, 1 week after virus administration, half of the animals switched to HFD and studied for further 16 weeks. Inclusion criteria were mice of a certain age (i.e. 9–12 weeks at the beginning of the experiment). Criteria for exclusion of mice from study groups were obvious infections/wounds which would impact on feeding behaviour as well as metabolic profile. These criteria were pre-established. Animal experiments were conducted according to local, national and EU ethical guidelines and approved by local regulatory authorities (Regierungspräsidium Karlsruhe, DEU) and conformed to ARRIVE guidelines.

### Metabolic phenotyping

For starvation experiments, mice were placed in fresh cages and food was withdrawn while maintain access to drinking water for 24 h from ZT1. For all GADD45 experiments, a control diet (Research diets D12450B, New Brunswick, USA) was used, and experiments were conducted after at least 1-week adaptation to the diet. To study diet-induced obesity, a high-fat diet was used (Research diets D12492, New Brunswick, USA). For comprehensive metabolic/behavioural phenotyping, the TSE Phenomaster system was used that permits automated and simultaneous monitoring of indirect calorimetry, body mass, food and water intake and 3D activity of individually housed mice (Tschop *et al*, 2012). Mice were acclimated to the system 4 days prior to the initiation of the experiments. For standard experiments, blood was taken after cervical dislocation and organs including liver, adipose tissue depots and gastrocnemius muscles were collected, weighed, snap-frozen in liquid nitrogen and stored at −80°C until further analysis.

To study the dynamics of systemic metabolism, we performed tolerance tests according to established guidelines (Ayala *et al*, 2010). In particular, we performed an intraperitoneal insulin (1 IU/kg; Huminsulin Normal, DEU) tolerance test in the 5–6 h fasted state

between ZT5-9. An oral lipid (100 µl/mouse olive oil delivered by oral gavage; O1514, Sigma-Aldrich, DEU) tolerance test was conducted in the overnight (i.e. 14–16 h) fasted state between ZT1-4. Furthermore, blood samples were collected from the tail vein in the 5–6 h fasted state (ZT6-7) for assessment of blood glucose and serum insulin levels for the calculation of HOMA-IR ((glucose (mM) × insulin (pM))/3857), which is a good surrogate measure of whole-body insulin action in mice (Lee *et al*, 2008). To assess tissue-specific insulin signalling, mice were fasted for 6 h, then insulin (Huminsulin, Lilly) was injected (10 mU/g body weight) and 15 min later, liver tissue was rapidly harvested and frozen in LN$_2$ following euthanisation by cervical dislocation (Agouni *et al*, 2010).

## Human subjects

Liver tissue samples were obtained from 37 Caucasian lean and obese men, 14 with and 23 without type 2 diabetes, who underwent open abdominal surgery for Roux-en-Y bypass, sleeve gastrectomy or elective cholecystectomy. Liver biopsy was taken during the surgery, immediately frozen in liquid nitrogen and stored at −80°C until further use. The phenotypic characterisation of the cohort has been performed as described previously (Kloting *et al*, 2010). Serum samples and liver biopsies were taken between 8 am and 10 am after an overnight fast. The study was approved by the local ethics committee of the University of Leipzig, Germany (363-10-13122010 and 017-12-230112). All patients gave preoperative written informed consent for the use of their samples.

## Blood metabolites & hormones

Blood glucose levels were determined using an automatic glucose monitor (One Touch, LifeScan). In addition, commercially available kits were used to measure serum non-esterified fatty acids (NEFA; NEFA-HR, Wako), glycerol/triglyceride (TG; TR-0100; Sigma-Aldrich), ketone bodies (KB; Autokit 3-HB, Wako), cholesterol (CH200, Randox) and insulin (80-INSMS-E01, Alpco) essentially according to manufacturer's instructions. All samples were loaded in order to fit within the assay range of the reagents supplied. Acyl-carnitines were determined in serum by electrospray ionisation tandem mass spectrometry (ESI-MS/MS) according to a modified method as previously described (Sauer *et al*, 2006), using a Quattro Ultima triple quadrupole mass spectrometer (Micromass, Manchester, UK) equipped with an electrospray ion source and a Micromass MassLynx data system.

## Tissue metabolite extraction and assay

For tissue lipid determinations, frozen tissue samples were pulverised, weighed and extracted. Lipid analyses were conducted according to established guidelines (Argmann *et al*, 2006) using glycerol/triglyceride (TR-0100; Sigma-Aldrich), NEFA (NEFA-HR, Wako) and cholesterol (CH200, Randox) assay kits. For liver glycogen determination, ~50 mg of liver powder was carefully weighed and extracted in 30% KOH followed by 70% ethanol precipitation. Resuspended glycogen was digested with amyloglucosidase (A7095, Sigma-Aldrich, DEU) and glycogen digests were then assayed using a glucose assay kit (GAHK20, Sigma-Aldrich, DEU). Liver tissue long-chain acyl-CoA species were extracted (Golovko & Murphy, 2004)

and measured using an enzymatic assay (Barber & Lands, 1971). Values were calculated as molar concentration per gram wet tissue.

## *Ex vivo* long-chain fatty acid metabolism

These experiments were carried out on precision-cut liver slices (de Graaf *et al*, 2010) according to established guidelines for long-chain fatty acid (LCFA) metabolism experiments (Watt *et al*, 2012). In particular, liver slices were taken and prepared (in absence of Insulin and Dexamethasone) from fed or fasted mice and pre-incubated in Williams media E containing 50 µg/ml gentamycin, 5% dialysed FBS, 5 mM D-glucose, 0.3 mM pyruvate, 0.1 µM methyl-linoleate and an amino acid mixture resembling the hepatic portal vein concentrations (Patti *et al*, 1998). After 1 h, slices were then incubated with the same media containing 0.5 mM carnitine and BSA-conjugated palmitic acid (50 µM) plus BSA-conjugated oleic and linoleic acid mix (150 µM; L9655, Sigma-Aldrich), representing the major (i.e. 67%) LCFA species in blood serum (Masood *et al*, 2005) together with trace amounts of [9,10-$^3$H(N)]-palmitic acid (ART0129; American Radiolabeled Chemicals, USA), to trace LCFA metabolism, and [1-$^{14}$C]-R-2-bromopalmitic acid (ARC3623; American Radiolabeled Chemicals, USA), a non-metabolisable palmitate analogue (Oakes *et al*, 1999) to trace LCFA uptake. Incubations were conducted for 3 h after which media was collected and slices were washed in ice-cold PBS, spot-dried and snap-frozen in LN$_2$. Liver slice samples were homogenised in Solvable™ (Perkin-Elmer, DEU) and media were extracted (twice) according to Folch method (Folch *et al*, 1957) in order to count only the aqueous $^3$H$_2$O reflecting FA oxidation. Samples were then analysed by dual-dpm counting (Packard 2200CA Tri-Carb Liquid Scintillation analyzer; Packard Instruments, USA) using multi-purpose scintillant (Rotiszint® eco plus, Carl-Roth, DEU), and LCFA metabolism was calculated based upon the media tracer:tracee ratio (dpm/mol). Non-oxidative LCFA disposal (NOFAD) rate was calculated as the difference between LCFA uptake and oxidation rates. In addition, liver slices from the same mice were allowed to incubate in media described above without glucose but with (BSA-NEFA) or without (fatty acid-free BSA vehicle) for 18 h and glucose concentration of the media was measured (GAHK20, Sigma-Aldrich, DEU) and subsequently glucose production rate was calculated.

## Plasmids, RNA interference and recombinant viruses

AAV encoding control or specific miRNAs under the control of a hepatocyte-specific promoter were established, purified and tittered as described previously (Graham *et al*, 2008; Rose *et al*, 2011). For miRNA experiments, oligonucleotides targeting mouse GADD45β (5′- GGCGGCCAAACTGATGAATGT -3′) and non-specific oligonucleotides (5′-AAATGTACTGCGCGTGGAGAC-3′) were cloned into pcDNA6.2-GW/EmGFP-miR ["BLOCK-iT™ PolII miR RNAi Expression Vector Kit" (Invitrogen, Darmstadt, DEU)]. For the overexpression of GADD45β in the murine liver AD virus were produced. Therefore, mGadd45b cDNA (GenBank: BC023815.1; Source Bioscience, UK) was subcloned into pENTR-FLAG vector and subsequently recombined with the pAD/BLOCK-IT™ DEST vector (Invitrogen, DEU). Linearised plasmid was subsequently transfected into HEK293A cells to amplify viruses, and these were subsequently

purified were purified by the caesium chloride method and dialysed against phosphate-buffered saline buffer containing 10% glycerol prior to animal injection.

### Tissue RNA/protein extraction and analysis

Liver cell fractionation was performed using a simplified method adapted from Jiang and colleagues (Liu *et al*, 2011). Liver organelle fractionation was conducted via density-based separation (Cox & Emili, 2006). RNA was extracted from tissues using Qiazol and cDNA synthesised using the First Strand cDNA synthesis kit (Fermentas, DEU). Quantitative PCR was conducted using Taqman master mix and Taqman primer-probe assays (Life Technologies, DEU). Tissue protein extraction and immunoblotting was performed using standard methods using GADD45β (sc-8776), FATP2 (sc-161311), FABP1 (sc-50380), HNF4a (sc-6556), BCKDE1A (sc-67200), NTCP (sc-98485) and ARG1 (sc-21050) (Santa-Cruz Biotechnology, DEU); LC3B (2275), pT39-S6K1 (9205), p-ERK (9101), p-p38 (9211), p-eIF2a (9721) and GRP78 (3183) (Cell Signaling Technologies, USA); CD36 (AF2519, RnD Systems, USA); and the housekeeping protein VCP (ab11433, Abcam, UK) antibodies. Immunoprecipitation was conducted using anti-FLAG (A2220, Sigma-Aldrich, DEU) and anti-HA (A2095, Sigma-Aldrich, DEU) agarose from tissue lysates using standard protocols.

### Study design criteria and statistical analyses

Based upon preliminary data showing the expected effect size of major outcome variables, a power analysis was conducted in order to determine the minimal number of animals to be used for each experiment. For genotype difference studies, offspring mice from Het × Het breedings were initially randomised to each experiment group. Afterwards, counterbalancing was done in order to realise equal sample sizes per experimental group. When conducting studies, the investigators were aware of which mouse was in which experimental group due to prior genotyping and allocation. However, the technical assistants involved in the studies were blinded.

Statistical analyses were performed using *t*-tests (two-sided), or 2-way analysis of variance (ANOVA) with or without repeated measures, where appropriate, with Holm–Sidak-adjusted post-tests. Nonparametric tests (e.g. Mann–Whitney–Wilcoxon) were conducted when data were not normally distributed. Correlation was determined using Spearman's correlation coefficient. All analyses were carried out with SigmaPlot v.12 software (Systat Software GmbH, DEU).

**Expanded View** for this article is available online.

### Acknowledgements

We thank Prof. Christof Niehrs (IMB, Mainz, DEU) and his colleagues Dr. Matthias Gierl and Dr. Andrea Schäfer for providing the Gadd45 mouse breeding pairs as well as experimental support and advice. Furthermore, we appreciate the experimental support of Emil Karaulanov (IMB, Mainz), Prachiti Narvekar, Astrid Wendler, Katharina Sowodniok, Yvonne Feuchter (A170, DKFZ) and Marcus Jabs (A270, DKFZ) for helpful advice regarding the LC-Acyl-CoA assay. This work was supported by grants from the Helmholtz Association (ICEMED, Cross-Program Topic Metabolic Dysfunction) and the Deutsche Forschungsgemeinschaft (He3260/4-2) to S.H.

**The paper explained**

**Problem**

A hallmark of obesity-driven T2D is insulin resistance, and thus "insulin sensitisation" has been an attractive strategy for treatment. However, insulin resistance likely represents a physiological feedback mechanism to actually retard the development of obesity-driven complications. Thus, alternative strategies are warranted, such as mild and intermittent activation of stress-responsive pathways that are pro-adaptive.

**Results**

Here, we show that "growth arrest and DNA damage-inducible" GADD45β as a dysregulated gene transcript during fasting in several models of metabolic dysfunction including ageing, obesity/pre-diabetes and type 2 diabetes, in both mice and humans. Using whole-body knockout mice as well as liver/hepatocyte-specific gain- and loss-of-function strategies, we revealed a role for liver GADD45β in the coordination of liver fatty acid uptake, through cytoplasmic retention of FABP1, ultimately impacting obesity-driven hyperglycaemia.

**Impact**

We identified liver GADD45β as a novel regulator of systemic and hepatic lipid metabolism. Our findings demonstrate the importance of fasting hepatic lipid metabolism in systemic metabolic control and provide insight into the development of new therapies for metabolic dysfunction.

## Author contributions

JF, AZ, TPS, OS, SC, KS, NV, RMdG, KN, MBD, AM and AJR performed the experiments and analysed the samples. JGO coordinated the blood serum acylcarnitine profiling. MB conducted the human studies and provided data thereof. JF, TPS and AJR analysed the data. SH & AJR co-directed the research project. AJR and JF wrote the manuscript. Dr. Adam J. Rose is the guarantor of this work and, as such, had full access to all the data in the study and takes responsibility for the integrity of the data and the accuracy of the data analysis.

## Conflict of interest

The authors declare that they have no conflict of interest.

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
