## [Review Process File · EMBO Molecular Medicine]

Fasting-induced liver GADD45 restrains hepatic fatty acid uptake and improves metabolic health.

Jessica Fuhrmeister, Annika Zota, Tjeerd P. Sijmonsma, Oksana Seibert, Şahika Cıngır, Kathrin Schmidt, Nicola Vallon, Roldan M. de Guia, Katharina Niopek, Mauricio Berriel Diaz, Adriano Maida, Matthias Blüher, Jürgen G. Okun, Stephan Herzig, Adam J. Rose

Corresponding authors: Adam Rose and Stephan Herzig, Joint Research Division Molecular Metabolic Control, German Cancer Research Center, Center for Molecular Biology, Heidelberg University and Heidelberg University Hospital, Heidelberg

Review timeline:

Submission date:	03 September 2015
Editorial Decision:	28 September 2015
Revision received:	08 March 2016
Editorial Decision:	23 March 2016
Revision received:	06 April 2016
Accepted:	08 April 2016

Transaction Report:

Editor: Roberto Buccione

1st Editorial Decision

28 September 2015

Thank you for the submission of your manuscript to EMBO Molecular Medicine. We have now heard back from the three Reviewers whom we asked to evaluate your manuscript.

As you will see the main concern raised is fundamental and shared by Reviewers 2 and 3. Although I will not dwell into much detail, I would like to highlight the main points.

Both Reviewers are concerned about the lack of mechanistic analysis to explain the potential role of GADD45b in lipid/glucose metabolism, and I agree that without such analysis, the interest and also potential medical relevance of the work is compromised.

Reviewer 2 lists a number of other items for your action, including a request to verify whether livers with increased GADD45b feature higher ER stress, and a few other issue related to the clarity and presentation of data.

Reviewer 3 also points to various critical issues, including the need for direct demonstration of the impairment of the hepatic insulin pathway and follow-up on AAV injected db/db mice to verify for weight loss. S/he also notes other methodological points that require clarification.

In conclusion, while publication of the paper cannot be considered at this stage, we are prepared to consider a substantially revised submission, with the understanding that the Reviewers' concerns must be fully addressed with additional experimental data where appropriate and that acceptance of the manuscript will entail a second round of review. The overall aim is to significantly upgrade the impact, significance and usefulness of the dataset, which of course are of paramount importance for our title.

I understand that if you do not have the required data available at least in part, to address the above, this might entail a significant amount of time and additional work, I would therefore understand if you chose to rather seek publication elsewhere at this stage. Should you do so, we would welcome a message to this effect.

Please note that it is EMBO Molecular Medicine policy to allow a single round of revision only and that, therefore, acceptance or rejection of the manuscript will depend on the completeness of your responses included in the next, final version of the manuscript.

As you know, EMBO Molecular Medicine has a "scooping protection" policy, whereby similar findings that are published by others during review or revision are not a criterion for rejection. However, I do ask you to get in touch with us after three months if you have not completed your revision, to update us on the status. Please also contact us as soon as possible if similar work is published elsewhere.

Please note that EMBO Molecular Medicine now requires a complete author checklist (<http://embomolmed.embopress.org/authorguide#editorial3>) to be submitted with all revised manuscripts. Provision of the author checklist is mandatory at revision stage; The checklist is designed to enhance and standardize reporting of key information in research papers and to support reanalysis and repetition of experiments by the community. The list covers key information for figure panels and captions and focuses on statistics, the reporting of reagents, animal models and human subject-derived data, as well as guidance to optimise data accessibility.

***** Reviewer's comments *****

Referee #1 (Comments on Novelty/Model System):

The models are quite reasonable and of considerable interest, especially since human material is included.

Referee #1 (Remarks):

These studies are of considerable interest but the description of the results will require extensive revision. Specifically, throughout the manuscript there are extremely vague descriptions ("dysregulation", is a common term) and non-standard terminology (e.g., "uncovered") that makes it almost impossible for the reader to understand what the conclusions are. Basically what the authors

have shown is that GADD45beta is induced by fasting in wild-type mice but less so in obese mice, and is lower in diabetic humans, and GADD45beta knockout mice exhibit impairments in metabolic regulation. The authors should clearly state the role of GADD45beta rather than using the ambiguous terms such as "dysregulation".

Referee #2 (Remarks):

The paper describes the role of GADD45b in the regulation of systemic nutrient regulation. The authors show that GADD45b is regulated in response to fasting both in liver as well as in other tissues and that this regulation seems to be impaired in obese/diabetic/aged animal models. Loss of Gadd45b leads to an increased lipid clearance, enhanced accumulation of hepatic TAGs and impaired glucose homeostasis. This is shown genetically in a global ko mouse as well as through oe and kd using an a liver specific AAV system. The data are very interesting as they identify a new regulator of hepatic fasting response that also affects systemic metabolism. Unfortunately the authors do not provide any mechanistical evidence and some other concerns need to be addressed.

1. From the data presented here I would presume that GADD45b is important for FFA import in the liver, without affecting metabolism of fatty acids. This would lead to the development of hepatic steatosis, hepatic insulin resistance and altered glucose and insulin homeostasis. There is very little evidence that GADD45b influences a lipid-glucose axis such as G3P and DHAP levels. Such statements should be removed without the proper evidence. This said, it would be good to get at least some evidence, whether the phenotype observed is due to alterations in lipid transporter expression, even if delineating the complete mechanism might be impossible. Also measuring hepatic insulin sensitivity in HFD fed Gadd45b mice or in db mice treated with an AAV for Gad45b would give additional insights into the mechanism.
2. The data presented in Fig. 1 is well known can be moved to supplements, since the db/db model has been extensively studied with regards to metabolic inflexibility.
3. The data in SFig. 3 should be moved to the main figures, because even though it is negative, it illustrates important aspects of the phenotype.
4. Is there any indication that the livers with increased GADD45b have increased ER stress, especially in light of the fact that Gadd45b is a stress regulated gene.
5. The single Wblot (Fig. 2M) does not reflect the mRNA data. To me the densitometry in the fed and db samples seems to measure background. Is there maybe additional stabilization of Gadd45b in the fasting state independent of the mRNA levels. Protein data from other models would help to argue this point.
6. The data presented in Fig.5 is quite important as it shows the partial rescue of the metabolic phenotype of db mice. This part should be emphasized and the part pertaining to Fig.1 which is well known should be reduced.

Referee #3 (Remarks):

Fuhrmeister et al. study the potential role of GADD45b in the metabolic adaptations to fasting using several in vivo and in vitro models and knock-out or overexpression strategies in control and obese/diabetic rodents. GADD45b is strongly expressed in the liver of fasted control mice but to a much lower extent in the liver of several models of obesity and diabetes. GADD45b total or liver

specific deletion leads to a decreased serum NEFA and increased liver TG during fasting whereas its overexpression reverses the phenotype. GADD45b deletion also alters insulin sensitivity. In human liver biopsies, GADD45b expression is lower in T2Diabetic subjects and GADD45b expression is inversely correlated with fasting TG. The authors thus suggest that GADD45b could have a role in glucose/lipid metabolism adaptations during fasting periods.

1. Although the observations are interesting, the mechanisms by which GADD45b can modulate hepatic lipid/glucose metabolism are not addressed. TG concentration in the liver results from the interaction of many different pathways, NEFA uptake, NEFA esterification, NEFA oxidation, VLDL export, NEFA de novo synthesis. As a starting point for more in depth studies, it would be interesting to have an overall view of genes modulated in the absence of GADD45b.
2. When comparing liver TG in normal and db/db mice during fasting, it must be emphasized that whereas in the fed state, their origin in db/db mice is a high lipogenic rate, in the fasted state, lipogenesis is blunted and hepatic TG both in control and obese mice originate from the overflow of NEFA from adipose tissue. It thus does not really reflect an inflexibility of lipid metabolism in db/db mice.
3. In the ex vivo experiments, a number of methodological details are lacking. It is stated that either palmitic or oleic and linoleic acid were used. But the reason for using these different mixtures is not clear and not mentioned in the figure legends. It is also stated that bromopalmitic acid was used but what for, LCFA uptake measurement ?
4. The authors measure LCFA oxidation using a labeled palmitate and state that in the db/db liver slices, the enhanced NEFA uptake is not accounted for by an increased oxidation. However, it is likely that in the hepatocytes from db/db mice in vitro, there is a flux of fatty acids originating from endogenous sources (steatosis) thus decreasing the intracellular ratio tracer/tracee and artificially decreasing the oxidation rate. It is otherwise difficult to explain such a difference since the authors state that the storage rate was not affected. The authors have then a strange formula stating that " hence, it is likely to be another fate and thus we examined glucose output ". The reader might understand that NEFA end up in glucose which is obviously a biochemical non sense. NEFA can indeed stimulate gluconeogenesis but through their oxidation products, acetyl-CoA as an allosteric activator of pyruvate carboxylase and reducing equivalents necessary for the reaction catalysed by glyceraldehyde 3P dehydrogenase.
5. In the experiments shown in figure 3, panels F,G,H,I, was the body weight of the control and KO mice similar ? The differences in blood glucose and insulin concentrations are not impressive (this is also true in figure 4 H, G). Globally a euglycemic hyperinsulinemic glucose clamp would be more adequate to evaluate the insulin sensitivity of these mice. In addition, additional experiments confirming an impairment of the hepatic insulin signalling pathway (IRS tyr phosphorylation, PKB/Akt ser phosphorylation ...) must be performed.
6. In figure 4G, a star is indicated for the KO AD-NC group. It must be removed when considering the statistical significance.
7. In the experiments related in figure 5 A, B and C, it is important to document whether the db/db mice have lost weight after the AAV injection. Due to their high feeding rate, they are much more sensitive to an alteration of the feeding behavior due to an external stress (here the AAV injection).

***** Reviewer's comments *****

Referee #1 (Comments on Novelty/Model System):

The models are quite reasonable and of considerable interest, especially since human material is included.

Referee #1 (Remarks):

These studies are of considerable interest but the description of the results will require extensive revision. Specifically, throughout the manuscript there are extremely vague descriptions ("dysregulation", is a common term) and non-standard terminology (e.g., "uncovered") that makes it almost impossible for the reader to understand what the conclusions are. Basically what the authors have shown is that GADD45beta is induced by fasting in wild-type mice but less so in obese mice, and is lower in diabetic humans, and GADD45beta knockout mice exhibit impairments in metabolic regulation. The authors should clearly state the role of GADD45beta rather than using the ambiguous terms such as "dysregulation".

We thank the reviewer for their appreciation of the novel findings and implications of our studies.

We apologise for the vague and unclear reporting of our conclusions drawn from our experiments. We have attempted to clarify this. In particular, from our new set of experiments we have now solidified our data set with a putative mechanism of how GADD45B works. This has made us more confident to make more definitive statements.

Referee #2 (Remarks):

The paper describes the role of GADD45b in the regulation of systemic nutrient regulation. The authors show that GADD45b is regulated in response to fasting both in liver as well as in other tissues and that this regulation seems to be impaired in obese/diabetic/aged animal models. Loss of Gadd45b leads to an increased lipid clearance, enhanced accumulation of hepatic TAGs and impaired glucose homeostasis. This is shown genetically in a global ko mouse as well as through oe and kd using an a liver specific AAV system. The data are very interesting as they identify a new regulator of hepatic fasting response that also affects systemic metabolism. Unfortunately the authors do not provide any mechanistical evidence and some other concerns need to be addressed.

We appreciate this reviewers succinct summary of our studies and acknowledgement of the interesting and novel findings of our work.

1. From the data presented here I would presume that GADD45b is important for FFA import in the liver, without affecting metabolism of fatty acids. This would lead to the development of hepatic steatosis, hepatic insulin resistance and altered glucose and insulin homeostasis. There is very little evidence that GADD45b influences a lipid-glucose axis such as G3P and DHAP levels. Such statements should be removed without the proper evidence. This said, it would be good to get at least some evidence, whether the phenotype observed is due to alterations in lipid transporter expression, even if delineating the complete mechanism might be impossible. Also measuring hepatic insulin sensitivity in HFD fed Gadd45b mice or in db mice treated with an AAV for Gad45b would give additional insights into the mechanism.

In the past months we have strived to find a molecular mechanism(s) linking GADD45B to lipid metabolism. Despite its described role as a factor controlling gene expression at epigenetic and transcriptional levels as well as via interacting with MAPK and autophagy signalling, we could not confirm that these events are linked to the phenotypes observed as there were no changes in liver FA metabolism gene expression levels nor changes in markers of signalling pathways. However, when we went a step further and examined fatty acid transport/binding protein localisation, we uncovered that fatty acid binding protein 1, an important protein involved in hepatocellular fatty acid transport and metabolism, was altered. We confirmed that this was due to GADD45B using several of our sample sets and could show that GADD45B was in a complex with FABP1 in the liver. Furthermore, we believe that we may have uncovered a potential link to the glucose metabolism phenotype with GADD45B loss in the accumulation of liver long-chain acyl-CoA.

Several past (PMID:7657026; PMID:9399959; PMID:15864350) and recent (PMID:25662011; PMID:22344295) studies have highlighted that there is a lack of a direct role for insulin on the liver to regulate hepatic glucose production. Consistent with this, we have examined multiple key insulin signalling phospho-proteins from our HFD studies (see attached) and we conducted a new study to examine insulin signalling phospho-proteins in db/db mice with GADD45B overexpression, and observed a lack of difference between study groups. Thus, we rather believe that the altered fatty acid flux and subsequent accumulation of LC-acyl-CoA in the liver is linked to the glucose/insulin phenotypes observed and have added a discussion point on this.

2. The data presented in Fig. 1 is well known can be moved to supplements, since the db/db model has been extensively studied with regards to metabolic inflexibility.

*Respectfully we disagree with this statement. While it is true that the concept of metabolic inflexibility in obesity/diabetes may be causal for eventual lack of metabolic control, most studies on db/db mice deal with differential metabolic control in the **fed** state and we believe that the lower lipid levels observed in the fasted state in our multiple models of metabolic dysfunction are worthy of highlighting in the first figure. This not only solidifies the basis of the entire manuscript but lays the foundation for the later use of the models for finding a consistent molecular signature to study.*

3. The data in SFig. 3 should be moved to the main figures, because even though it is negative, it illustrates important aspects of the phenotype.

This was a good suggestion. We have moved the indirect calorimetry data as well as some of the metabolite data to the main figure.

4. Is there any indication that the livers with increased GADD45b have increased ER stress, especially in light of the fact that Gadd45b is a stress regulated gene.

We have measured two readouts of different arms of ER stress signalling; eIF2a phosphorylation and GRP78 expression in GADD45B KO mice in the fasted state and observed no differences when compared with wildtype mice. These results are included in SF8.

5. The single Wblot (Fig. 2M) does not reflect the mRNA data. To me the densitometry in the fed and db samples seems to measure background. Is there maybe additional stabilization of Gadd45b in the

fasting state independent of the mRNA levels. Protein data from other models would help to argue this point.

This might be due to threshold effects of mRNA transcription-translation. From qPCR analyses, the CT values are ~30 in the fed state and ~24-6 in the fasted state. Nevertheless, we have also performed additional immunoblots of the other mouse models (i.e. NZO and aged) and observe the same trend. These results have solidified the GADD45B protein expression data and are included in Figure 2.

6. The data presented in Fig.5 is quite important as it shows the partial rescue of the metabolic phenotype of db mice. This part should be emphasized and the part pertaining to Fig.1 which is well known should be reduced.

We agree that the data are important as they relate back to the initial basis of the study. We have used these samples again in figure 6 which demonstrate that GADD45B restoration in db/db mice results in a lowering of LC-acyl-CoA levels in the liver. This helps to bring the story in a full circle.

Referee #3 (Remarks):

Fuhrmeister et al. study the potential role of GADD45b in the metabolic adaptations to fasting using several in vivo and in vitro models and knock-out or overexpression strategies in control and obese/diabetic rodents. GADD45b is strongly expressed in the liver of fasted control mice but to a much lower extent in the liver of several models of obesity and diabetes. GADD45b total or liver specific deletion leads to a decreased serum NEFA and increased liver TG during fasting whereas its overexpression reverses the phenotype. GADD45b deletion also alters insulin sensitivity. In human liver biopsies, GADD45b expression is lower in T2Diabetic subjects and GADD45b expression is inversely correlated with fasting TG. The authors thus suggest that GADD45b could have a role in glucose/lipid metabolism adaptations during fasting periods.

We appreciate this thorough summary of our studies.

1. Although the observations are interesting, the mechanisms by which GADD45b can modulate hepatic lipid/glucose metabolism are not addressed. TG concentration in the liver results from the interaction of many different pathways, NEFA uptake, NEFA esterification, NEFA oxidation, VLDL export, NEFA de novo synthesis. As a starting point for more in depth studies, it would be interesting to have an overall view of genes modulated in the absence of GADD45b.

While we have not conducted a transcriptome screen of liver from our studies, we have conducted a focussed mRNA expression profiling of key genes in liver FA metabolism. While this revealed no differences we have observed an altered localisation of FABP1 in the livers of GADD45B KO mice

2. When comparing liver TG in normal and db/db mice during fasting, it must be emphasized that whereas in the fed state, their origin in db/db mice is a high lipogenic rate, in the fasted state, lipogenesis is blunted and hepatic TG both in control and obese mice originate from the overflow of NEFA from adipose tissue. It thus does not really reflect an inflexibility of lipid metabolism in db/db mice.

We agree with this and have removed this data.

3. In the ex vivo experiments, a number of methodological details are lacking. It is stated that either palmitic or oleic and linoleic acid were used. But the reason for using these different mixtures is not clear and not mentioned in the figure legends. It is also stated that bromopalmitic acid was used but what for, LCFA uptake measurement ?

We made a mistake in the prior methods section and appreciate this being brought to our attention. All three fatty acids were used. This is based upon recommendations and that these three FAs are the major FAs in blood. The methods section has also been altered to indicate why each tracer was used.

4. The authors measure LCFA oxidation using a labeled palmitate and state that in the db/db liver slices, the enhanced NEFA uptake is not accounted for by an increased oxidation. However, it is likely that in the hepatocytes from db/db mice in vitro, there is a flux of fatty acids originating from endogenous sources (steatosis) thus decreasing the intracellular ratio tracer/tracee and artificially decreasing the oxidation rate. It is otherwise difficult to explain such a difference since the authors state that the storage rate was not affected. The authors have then a strange formula stating that " hence, it is likely to be another fate and thus we examined glucose output ". The reader might understand that NEFA end up in glucose which is obviously a biochemical non sense. NEFA can indeed stimulate gluconeogenesis but through their oxidation products, acetyl-CoA as an allosteric activator of pyruvate carboxylase and reducing equivalents necessary for the reaction catalysed by glyceraldehyde 3P dehydrogenase.

It is true that we cannot completely explain the other "fate" of the FA metabolism in the db/db mouse livers and that the assumed enhanced lipolysis/turnover may have diluted the intracellular tracer:tracee ratio however, we don't have a good idea for an experiment to test this other than to use an inhibitor of lipolysis which would then probably affect other aspects of FA turnover indirectly anyway. Nevertheless, we have observed that long chain acyl-CoAs accumulate in liver tissue of both GADD45B KO and db/db mice highlighting that there is enhanced supply of FAs relative to their removal intracellularly.

Further, we agree that it biochemical non-sense that fatty acids can be converted to glucose due to enzymatic energetic constraints and apologise for the misleading/ambiguous sentence and have modified it accordingly.

5. In the experiments shown in figure 3, panels F,G,H,I, was the body weight of the control and KO mice similar ? The differences in blood glucose and insulin concentrations are not impressive (this is also true in figure 4 H, G). Globally a euglycemic hyperinsulinemic glucose clamp would be more adequate to evaluate the insulin sensitivity of these mice. In addition, additional experiments confirming an impairment of the hepatic insulin signalling pathway (IRS tyr phosphorylation, PKB/Akt ser phosphorylation ...) must be performed.

Yes, the body weights of WT and KO mice were similar, particularly on HFD where we revealed the slight differences (Fig. S5D). We agree that the effects on blood glucose are no impressive but we believe that the effects on insulin are substantial (~40% higher in KOs) and consistent and therefore robust. From our experience blood glucose is not affected greatly with HFD, but serum insulin is due to pancreatic adaptation to keep glucose in check, and thus we believe that there is impaired glucose metabolism in these mice upon chronic HFD feeding. Furthermore, the HOMA index has been previously compared against the clamp technique, with good correlations (see methods). Thus we do not think that it is necessary to conduct extra studies and perform the euglycemic hyperinsulinemic glucose clamp technique, which would be beyond the scope of the current manuscript.

Several past (PMID:7657026; PMID:9399959; PMID:15864350) and recent (PMID:25662011; PMID:22344295) studies have highlighted that there is a lack of a direct role for insulin on the liver to regulate hepatic glucose production. Consistent with this, we have examined multiple key insulin signalling phospho-proteins from our HFD studies (see attached) and we conducted a new study to examine insulin signalling phospho-proteins in db/db mice with GADD45B overexpression, and observed a lack of difference between study groups. Thus, we rather believe that the altered fatty acid flux and subsequent accumulation of LC-acyl-CoA in the liver is linked to the glucose/insulin phenotypes observed and have added a discussion point on this.

6. In figure 4G, a star is indicated for the KO AD-NC group. It must be removed when considering the statistical significance.

The star has been removed.

7. In the experiments related in figure 5 A, B and C, it is important to document whether the db/db mice have lost weight after the AAV injection. Due to their high feeding rate, they are much more sensitive to an alteration of the feeding behaviour due to an external stress (here the AAV injection).

This is a good point. We have now included data on the body mass change of the experiment and show no differences between experimental groups (Fig. FS7).

Summary insulin signaling

Thank you for the submission of your revised manuscript to EMBO Molecular Medicine. We have now received the enclosed reports from the referees that were asked to re-assess it. The reviewers are now globally supportive and I am pleased to inform you that we will be able to accept your manuscript pending the following final amendments:

- 1) As you will see, Reviewer 2, while recognising the significant improvement, has a few remaining concerns for you to deal with. On one hand, s/he would like you to discuss the fact that Gadd45b appears to act via FABP1. On the other, s/he feels that some important experimental details are missing. I would ask you to please comply with these final requests. Depending on the completeness of your reply, I may make an editorial decision on your manuscript.
- 2) As per our Author Guidelines, the description of all reported data that includes statistical testing must state the name of the statistical test used to generate error bars and P values, the number (n) of independent experiments underlying each data point (not replicate measures of one sample), and the actual P value for each test (not merely 'significant' or 'P < 0.05'). If necessary or preferred, you may add an additional appendix table to list all the P values, in which case, please make sure the manuscript is modified accordingly with the appropriate callouts!
- 3) We are now encouraging the publication of source data, particularly for electrophoretic gels and blots, with the aim of making primary data more accessible and transparent to the reader. Would you be willing to provide a PDF file per figure that contains the original, uncropped and unprocessed scans of all or at least the key gels used in the manuscript? The PDF files should be labeled with the appropriate figure/panel number, and should have molecular weight markers; further annotation may be useful but is not essential. The PDF files will be published online with the article as supplementary "Source Data" files. If you have any questions regarding this just contact me.
- 4) Please change your Supplementary Figures file to Appendix (Level 3 - please refer to our Author Guidelines) and adjust the manuscript callouts accordingly
- 5) Please include "The Paper Explained" section in the manuscript

Please submit your revised manuscript within two weeks. I look forward to seeing a revised form of your manuscript as soon as possible.

***** Reviewer's comments *****

Referee #2 (Remarks):

The paper analyzes the role of Gadd45b in hepatic function. Based on the observation that this gene is highly regulated upon fasting in liver the authors study both global ko animals as well as virus induced kd and overexpression. Gadd45 loss leads to improved lipid clearance and hepatic lipid accumulation. And Gadd45b overexpression also in diabetic models seems to partially restore liver function. The revised version is very much improved and in my opinion only a few minor points are missing:

The mechanistic data is interesting and clearly demonstrates that the published function of Gadd45b does not seem to play a role in this context. Interestingly Gadd45b seems to act through FABP1, which has previously been shown to regulate fatty acid import into hepatocytes, a fact that should be better addressed in the discussion as it links the observations of the paper with a possible mechanism. Nevertheless, a few controls are missing and should be provided. First of all, marker genes to demonstrate the efficacy of separation in 6C should be provided. Second, the cellular distribution of Gadd45b should be shown in the same blot to understand where Gadd45b and Fabp1

might interact.

2nd Revision - authors' response

06 April 2016

Referee #2 (Remarks):

The paper analyzes the role of Gadd45b in hepatic function. Based on the observation that this gene is highly regulated upon fasting in liver the authors study both global ko animals as well as virus induced kd and overexpression. Gadd45 loss leads to improved lipid clearance and hepatic lipid accumulation. And Gadd45b overexpression also in diabetic models seems to partially restore liver function. The revised version is very much improved and in my opinion only a few minor points are missing:

We thank this reviewer for their time and efforts and for their succinct and precise summary of our studies. We appreciate the sentiment that our manuscript is much improved since the prior version.

The mechanistic data is interesting and clearly demonstrates that the published function of Gadd45b does not seem to play a role in this context. Interestingly Gadd45b seems to act through FABP1, which has previously been shown to regulate fatty acid import into hepatocytes, a fact that should be better addressed in the discussion as it links the observations of the paper with a possible mechanism. Nevertheless, a few controls are missing and should be provided. First of all, marker genes to demonstrate the efficacy of separation in 6C should be provided. Second, the cellular distribution of Gadd45b should be shown in the same blot to understand where Gadd45b and Fabp1 might interact.

We appreciate these comments and we have amended the manuscript accordingly which has improved the quality by the addition of appropriate controls for the qualification of the methods used. For the fractionation work, marker protein expression has now been conducted and is included in Figure EV8, with the Gadd45B expression shown in the main figure (i.e. Figure 6C), which clearly shows enrichment of GADD45B in the cytosolic fraction, of which FABP1 is also enriched. In addition, while in review we conducted further experiments showing mislocalisation of FABP1 expression in the liver of obese/diabetic mice, which could be reversed by GADD45B overexpression (Figure 6F). Taken together, we believe that these additional experiments have strengthened the conclusions made that GADD45B acts via cytosolic FABP1 binding and retention, and thereby the overall quality of the manuscript. In addition, we have expanded our discussion section on the possible role of FABP1 localisation contributing to metabolic phenotypes observed.

Corresponding Author Name: Adam Rose

Manuscript Number: